# Polynomial filter diagonalization of large Floquet unitary operators

David J. Luitz

Max Planck Institute for the Physics of Complex Systems,
Noethnitzer Str. 38, 01167 Dresden, Germany
dluitz@pks.mpg.de

July 21, 2021

## Abstract

Periodically driven quantum many-body systems play a central role for our understanding of nonequilibrium phenomena. For studies of quantum chaos, thermalization, many-body localization and time crystals, the properties of eigenvectors and eigenvalues of the unitary evolution operator, and their scaling with physical system size $L$ are of interest. While for static systems, powerful methods for the partial diagonalization of the Hamiltonian were developed, the unitary eigenproblem remains daunting.

In this paper, we introduce a Krylov space diagonalization method to obtain exact eigenpairs of the unitary Floquet operator with eigenvalue closest to a target on the unit circle. Our method is based on a complex polynomial spectral transformation given by the geometric sum, leading to rapid convergence of the Arnoldi algorithm. We demonstrate that our method is much more efficient than the shift invert method in terms of both runtime and memory requirements, pushing the accessible system sizes to the realm of 20 qubits, with Hilbert space dimensions $\geq 10^6$.

# 1 Introduction

Periodically driven, Floquet quantum many-body systems host fascinating nonequilibrium phenomena [1, 2]. While they generically relax to a featureless state [3, 4] independent of the initial state, they can undergo nonequilibrium phase transitions, such as the many-body localization transition [5–10]. In this context, Floquet systems are often cleaner counterparts of Hamiltonian systems, capturing the essence of these phenomenona, due to the absence of an energy structure and a uniform density of states. This is for example useful for high quality tests of the eigenstate thermalization hypothesis [4, 11–17].

Interestingly, Floquet quantum many-body systems can exhibit altogether new physics, absent in Hamiltonian systems [18], such as robust [19–24] or fine tuned [25, 26] *time crystals* or prethermal states [27–32].

Many questions, in particular in the context of the many-body localization transition, rely on studying the scaling of the properties of eigenvectors of the unitary one period evolution operator $U$ with system size. While in static systems powerful methods like shift-invert diagonalization [33,34] and polynomial filter diagonalization [35–38] were developed for finding interior eigenpairs of a large hermitian and sparse Hamiltonian up to Hilbert space dimensions of $10^7$, the case of the dense unitary eigenproblem remains daunting.

Although progress was made for the special case deep in the many-body localized phase based on matrix product state variants of the shift invert technique [39, 40] for the Floquet operator [41], a general purpose method which can go beyond full diagonalization of the unitary matrix $U$, limited to system sizes of about $L = 14 \ldots 16$ qubits is yet missing.

In this paper, we introduce a new method using a spectral transformation given by the geometric sum $g_k(U)$ of order $k$. The spectral transformation is a complex polynomial of $U$ and an efficient matrix vector product $g_k(U)|\psi\rangle$ can be defined, provided there is an efficient matrix vector product $U|\psi\rangle$. This is the case in local Floquet systems (e.g. in a matrix product operator formulation). The spectral transformation is designed to enhance the absolute value of eigenvalues of $U$ close to an arbitrary target on the unit circle, and to reduce the absolute value of all other eigenvalues, thus allowing rapid convergence of the Arnoldi algorithm to the requested eigenpairs of $g_k(U)$.

We show that this procedure is effective and can be carried out with a low memory footprint, compared to dense shift-invert or full diagonalization. Effectively, it gives access to system sizes up to $L \geq 20$ qubits, while full diagonalization is limited to $L \approx 15$.

This computational advantage makes extensive finite size scaling studies of Floquet MBL systems possible, and can help to make progress on the recent debate on the stability of many-body localization in the thermodynamic limit [42–49], which highlights the importance of finite size effects.

# 2 Model

To investigate the performance of our method, we consider a simple generic model for a time periodic one dimensional quantum many-body system of $L$ qubits, with a Hilbert space of dimension $d = 2^L$. The model is designed such that it is ergodic, with highly entangled eigenstates, and is not tractable by alternative methods, e.g. tensor network techniques [41].

The evolution operator $U$ over one driving period is given by a two layer brickwork circuit,

composed of two site unitaries.

$$U = \qquad\qquad\qquad\qquad\qquad\qquad\qquad\qquad (1)$$

Each of the boxes in Eq. (1) represents a unitary $u_{i,i+1} \in \mathbb{C}^{4\times4}$, acting on qubit $i$ and its right neighbor $(i+1)$. At the boundaries, we fill in single qubit unitaries $u_1, u_L \in \mathbb{C}^{2\times2}$ if needed. We sample all unitaries randomly from the uniform measure on the unitary group [50]. Our construction ensures that each link in the chain of qubits is represented by a $4 \times 4$ unitary, corresponding to generic 2-qubit interactions.

We can express $U$ in terms of the two layers $U_a$ and $U_b$ (for even $L$):

$$
\begin{aligned}
U &= U_a U_b, \\
U_a &= u_{1,2} \times u_{3,4} \times \cdots \times u_{L-1,L} \\
U_b &= u_1 \times u_{2,3} \times \cdots \times u_{L-2,L-1} \times u_L.
\end{aligned}
\qquad (2)
$$

It is important to note that the unitary matrix $U$ is *dense* in the computational basis. To construct an efficient matrix vector product $|\psi'\rangle \leftarrow U|\psi\rangle$, we split the circuit in a left part $U_{\mathrm{L}} \in \mathbb{C}^{d_{\mathrm{L}} \times d_{\mathrm{L}}}$ (red tensors in Eq. (1)) and a right part $U_{\mathrm{R}} \in \mathbb{C}^{d_{\mathrm{R}} \times d_{\mathrm{R}}}$ (blue tensors in Eq. (1)). It is advisable to chose $d_{\mathrm{L}}, d_{\mathrm{R}} \approx \sqrt{2^L}$.

The Floquet operator is then decomposed into $U = (U_{\mathrm{L}} \times \mathbb{1})(\mathbb{1} \times U_{\mathrm{R}})$, and we can calculate $U|\psi\rangle$ by two matrix products, first calculating

$$
\begin{aligned}
\psi_{d_{\mathrm{L}} \times d/d_{\mathrm{L}}} &\leftarrow U_{\mathrm{L}} \psi_{d_{\mathrm{L}} \times d/d_{\mathrm{L}}} \quad \text{and then} \\
\psi_{d/d_{\mathrm{R}} \times d_{\mathrm{R}}} &\leftarrow \psi_{d/d_{\mathrm{R}} \times d_{\mathrm{R}}} U_{\mathrm{R}}^T.
\end{aligned}
\qquad (3)
$$

Note, that here $\psi_{d_{\mathrm{L}} \times d/d_{\mathrm{L}}}$ and $\psi_{d/d_{\mathrm{R}} \times d_{\mathrm{R}}}$ are two *different* reshapings of the vector $|\psi\rangle$ into matrices of dimensions indicated in the subscript.

The advantage of this procedure is that instead of the very large matrix $U \in \mathbb{C}^{d\times d}$, we only need to store two much smaller matrices $U_{\mathrm{L}}$ and $U_{\mathrm{R}}^T$, and we have expressed the matrix vector product in terms of two matrix products, with efficient memory access per floating point operation.

This decomposition is specific to brickwork circuits, but for general one dimensional local Floquet problems efficient matrix free matrix vector products can be formulated, for example based on a matrix product operator formulation of $U$ [41]. The matrix product operator is guaranteed by locality to have a constant bond dimension due to the area law of the operator entanglement entropy of $U$ [51, 52].

## 3   Method

Our goal is to calculate a subset of the eigenpairs $\{\omega_n, |n\rangle\}$ of the large unitary matrix $U$ in such a way that we obtain *all* $n_{\mathrm{ev}}$ eigenpairs with eigenvalue $\omega_n$ closest to a target $z_{\mathrm{tgt}} \in \mathbb{C}$. The eigenvalues $\omega_n$ of $U$ lie on the complex unit circle, $|\omega_n| = 1$ and can therefore not be separated by magnitude, which is necessary for the convergence of Krylov space diagonalization techniques.

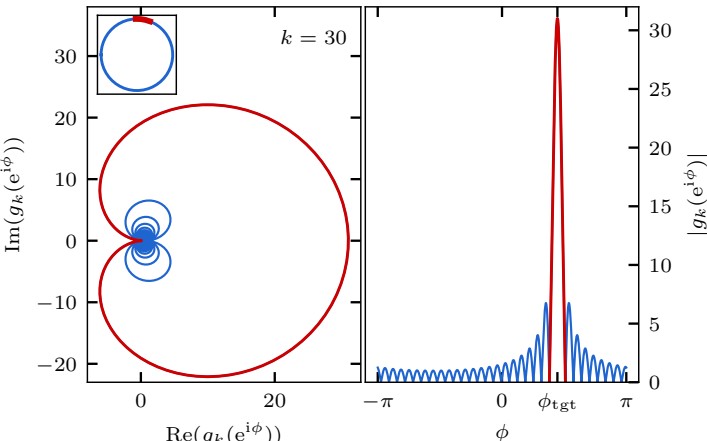

Figure 1: Left: Mapping of the complex unit circle $|z| = 1$ (inset) by the geometric sum $g_k(z)$ with polynomial order $k = 30$. The "target" arc of the unit circle highlighted in red is mapped to the red line in the main panel by $g_k(z)$. Right: Absolute value of the mapped values as a function of phase angle $\phi$. The part of the curve marked in red is the mapped "target" arc of the unit circle, given by the target phase $\phi_{\text{tgt}}$.

To achieve this, *spectral transformations* $f(U)$ can be used to transform the eigenvalues $\omega_n \to f(\omega_n)$, while leaving the the eigenvectors $|n\rangle$ invariant. If the magnitude $|f(\omega_n)|$ is large for eigenvalues $\omega_n$ close to the target and small otherwise, e.g. the Arnoldi algorithm can be used to calculate the eigenpairs of interest of $f(U)$, while only the matrix vector product $|\psi'\rangle \leftarrow f(U)|\psi\rangle$ is needed.

One of the most effective spectral transformations is the "shift and invert" transformation

$$f_{\text{sinvert}}(U) = (U - z_{\text{tgt}}\mathbb{1})^{-1} \tag{4}$$

which provides excellent convergence of the Arnoldi algorithm if the target is chosen on or close to the unit circle. The downside of $f_{\text{sinvert}}$ is that for the matrix vector product $(U - z_{\text{tgt}}\mathbb{1})^{-1}|\psi\rangle$, an inversion is involved. Due to the dense spectrum of $U$, the condition number of $U - z_{\text{tgt}}\mathbb{1}$ is exponentially large in system size and therefore only a direct solution using a $LU$ decomposition of the dense matrix $U - z_{\text{tgt}}\mathbb{1}$ can be used, with a memory complexity $O(d^2)$ and runtime complexity $O(d^3)$. We provide benchmark results of this technique in Tab. 1 for comparison.

Polynomial spectral transformations are useful, because they do not suffer from large memory requirements since any power of $U$ can be applied to a vector $|\psi\rangle$ by repeated matrix vector products $U^k|\psi\rangle = U(U \ldots (U|\psi\rangle))$. Generally, the polynomial of degree $k$

$$p_k(U) = \sum_{m=0}^{k} a_m U^m \tag{5}$$

can be efficiently multiplied onto a vector to obtain $p_k(U)|\psi\rangle$.

We argue here that an effective complex polynomial spectral transformation to single out eigenpairs with eigenvalue closest to $z_{\text{tgt}} = e^{i\phi_{\text{tgt}}}$ is given by the geometric sum

$$g_k(U) = \sum_{m=0}^{k} e^{-im\phi_{\text{tgt}}} U^m. \tag{6}$$

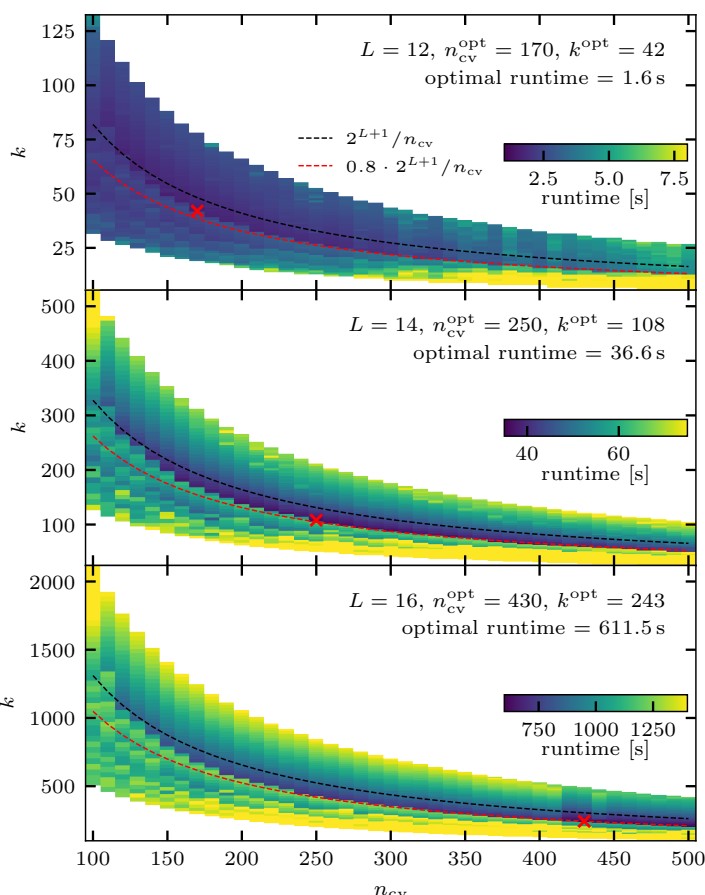

Figure 2: Benchmark calculation of $n_{\mathrm{ev}} = 50$ eigenpairs closest to $z_{\mathrm{tgt}} = 1$ using 4 cores of an AMD EPYC 7H12 2.6 GHz CPU. The colormap shows the measured runtimes in seconds, the black dashed line indicates Krylov space dimensions equal to the number of eigenvalues on the outer apple line in Fig. 1. The red dashed line shows the Krylov space dimension 80% smaller than this value. Red crosses show the position of absolute minimal runtime. Scanning algorithmic parameters $n_{\mathrm{cv}}$ and $k$ disabled core boost and therefore numbers can differ from Tab. 1.

This polynomial maximizes the absolute value of eigenvalues close to the target and has minimal modulus outside the target region. We conjecture here, that it is in fact the optimal choice, although we have checked this only numerically. The phase factor can be understood by noticing that multiplication by $\mathrm{e}^{-\mathrm{i}\phi_{\mathrm{tgt}}}$ rotates the eigenvalues of $U$ closest to $\mathrm{e}^{\mathrm{i}\phi_{\mathrm{tgt}}}$ to the proximity of 1.

The geometric sum $g_k(U)$ has the closed form (if $\omega_n \neq 1$)

$$g_k(U) = \left[\mathbb{1} - \mathrm{e}^{-\mathrm{i}(k+1)\phi_{\mathrm{tgt}}} U^{k+1}\right] \left[\mathbb{1} - \mathrm{e}^{-\mathrm{i}\phi_{\mathrm{tgt}}} U\right]^{-1}, \tag{7}$$

and has some similarity with $f_{\mathrm{sinvert}}$.

Fig. 1 illustrates the mapping of the unit circle by $g_k(z)$. A number $\mathrm{e}^{\mathrm{i}\phi}$ on the unit circle is mapped onto

$$g_k(\mathrm{e}^{\mathrm{i}\phi}) = \frac{1 - \mathrm{e}^{\mathrm{i}(k+1)(\phi-\phi_{\mathrm{tgt}})}}{1 - \mathrm{e}^{\mathrm{i}(\phi-\phi_{\mathrm{tgt}})}}. \tag{8}$$

The left panel shows the transformed unit circle under $g_k(z)$ on the complex plane, while the right panel depicts $|g_k(e^{i\phi})|$ as a function of the phase angle $\phi$. In the limit $\phi \to \phi_{\text{tgt}}$, $g_k(e^{i\phi})$ is on the real axis and given by $k+1$. This is the place with maximal modulus. If we tune $\phi$ away from $\phi_{\text{tgt}}$, $g_k(e^{i\phi})$ moves away from this point, and the modulus decreases, until we reach $g_k(e^{i\phi}) = 0$ for $\phi - \phi_{\text{tgt}} = \pm\frac{2\pi}{k+1}$. If $\phi_{\text{tgt}} - \frac{2\pi}{k+1} \le \phi \le \phi_{\text{tgt}} + \frac{2\pi}{k+1}$, we therefore get the "apple" shaped outer line in the left panel of Fig. 1, while the rest of the circle is compressed into the inner spirals. The target arc satisfying this condition is shown in red in Fig. 1. It is noteworthy that the target arc is not only strongly enhanced in magnitude by $g_k$, but also shows the largest separation of eigenvalues on the complex plane. These features are important for a rapid convergence of the Arnoldi algorithm.

Quantum many-body Floquet systems typically have a uniform eigenvalue density on the unit cirle. Therefore, on average for a fixed polynomial order $k$, the $2d/(k+1)$ eigenvalues closest to $z_{\text{tgt}}$ will be mapped to the outer "apple" line by $g_k$.

## 4   Arnoldi algorithm for $g_k(U)$

The Arnoldi algorithm [53] is a generalization of the Lanczos method [54] to nonhermitian matrices. It is a numerically stable variant of the power iteration, and iteratively builds an orthonormal basis $\{v_j\}$ of the Krylov space $\text{span}\left(|\psi\rangle, g_k(U)|\psi\rangle, g_k^2(U)|\psi\rangle \ldots g_k^{n_{\text{cv}}-1}|\psi\rangle\right)$ of dimension $n_{\text{cv}}$, starting from a random initial vector $|\psi\rangle$. Raising $g_k$ to high powers in this process filters out components of eigenvectors of $g_k$ which are in the target space (i.e. whose eigenvalues of $g_k$ have a large magnitude and are therefore enhanced). At the same time, $g_k(U)$ is projected into the Krylov space by the matrix $V \in \mathbb{C}^{d \times n_{\text{cv}}}$ with columns given by $v_j$, yielding the upper Hessenberg matrix

$$H_m = V^\dagger g_k(U) V. \tag{9}$$

The eigenvalues $\lambda_i$ and eigenvectors $x_i$ of $H_m$ yield the Ritz pairs $(\lambda_i, V x_i)$, which are approximations of the eigenpairs of $g_k(U)$. If the dimension of the Krylov space $n_{\text{cv}}$ is sufficiently large, the Ritz pairs will converge with the number of Arnoldi iterations. Typically, $\lambda_i$ with the largest magnitude converge first, and therefore an effective spectral transformation is important. One should not expect to converge all $n_{\text{cv}}$ Ritz pairs, but rather chose a maximal size of the Krylov space $n_{\text{cv}} > n_{\text{ev}}$, if $n_{\text{ev}}$ eigenvalues are required.

Once $n_{\text{ev}}$ Ritz pairs are converged, we obtain excellent approximations for eigenpairs of $g_k(U)$. The eigenvectors $|n\rangle$ of $g_k(U)$ are also eigenvectors of $U$. The eigenvalues $\lambda_n$ of $g_k(U)$ are related to the corresponding eigenvalues $\omega_n$ via $g_k(\omega_n) = \lambda_n$. Rather than solving this equation by root finding, we calculate them from $\omega_n = \langle n|U|n\rangle$. This has the advantage that we can at the same time check the residuals [1]

$$r_n = \| U|n\rangle - \omega_n|n\rangle \|_2 \tag{10}$$

as a measure of eigenpair quality.

---

[1] We note here in passing that for very high orders of filtering polynomials the numerical precision of applying $g_k(U)$ may be insufficient. In this case, one can perform one iteration of the Ritz method by diagonalizing $A_{nm} = \langle \tilde{n}|U|\tilde{m}\rangle$ with approximate eigenvectors $|\tilde{n}\rangle$ of $g_k(U)$ to improve the precision of Ritz pairs of $U$. We did however not observe such problems in practice.

We pursue here the following strategy: Since the eigenvalues on the outer "apple" line of $g_k(z)$ are well separated from the rest of the spectrum, it is advisable to use a Krylov space dimension $n_{\mathrm{cv}} = 2d/(k+1)$ (in practice, one can take it about 80% smaller as we will see below). If we want to calculate $n_{\mathrm{ev}}$ eigenpairs, we furthermore use $n_{\mathrm{cv}} > 2n_{\mathrm{ev}}$. We use the reference implementation of the implicitly restarted Arnoldi method as provided by `arpack` [55].

For large system sizes, the runtime of the algorithm is dominated by matrix vector products $g_k(U)|\psi\rangle$. We can therefore estimate the computational cost for the matrix vector product proposed in Eq. (3). A single matrix vector product, rephrased as two matrix matrix products of matrices of dimension $2^{L/2}$ has a cost of $O(2^{3L/2})$. For a fixed number of required eigenpairs $n_{\mathrm{ev}}$, we use an exponentially large polynomial order $k = 0.8 \cdot 2^{L+1}/n_{\mathrm{cv}}$, and therefore a single matrix vector product $g_k(U)|\psi\rangle$, requiring $k$ matrix vector products involving $U$, has an asymptotic cost of $O(2^{5L/2}/n_{\mathrm{cv}})$. In the Arnoldi algorithm, we need at least $n_{\mathrm{cv}}$ of these matrix vector products, and hence the overall asymptotic runtime complexity is $O(2^{2.5L})$. We note that in the benchmarks in Sec. 5 this asymptotic complexity is only approximately visible, since even for the largest system sizes, CPU specific effects like cache sizes play a role.

## 5 Benchmarks

To find the optimal algorithmic parameters $k$ and $n_{\mathrm{cv}}$, we perform benchmark calculations to obtain $n_{\mathrm{ev}} = 50$ eigenpairs closest to $z_{\mathrm{tgt}} = 1$ of Floquet random circuits from Eq. (1) of length $L = 12, 14, 16$ for a range of Krylov space dimensions $n_{\mathrm{cv}}$ and polynomial orders $k$. The results are shown in Fig. 2. The black dashed line corresponds to the $n_{\mathrm{cv}} = 2^{L+1}/k$, where the Krylov space dimension is equal to the number of eigenvalues on the outer apple line in Fig. 1. The colormap shows the obtained runtimes in seconds. There is a distinct minimum visible in the runtime, when the Krylov space dimension is about $0.8 \cdot 2^{L+1}/k$ (red dashed) for large $n_{\mathrm{cv}}$. This corresponds to the number of eigenvalues with larger modulus than the rest of the spectrum and is therefore the optimal size of the Krylov space for each $k$. We observe a tendency that for larger sizes, larger Krylov space dimensions are better. And we are therfore using $n_{\mathrm{cv}} = \lfloor 2^{L/2+1} \rfloor$, $k = 0.8 \cdot 2^{L+1}/n_{\mathrm{cv}}$ in the following benchmarks for larger sizes.

For an assessment of the performance of our method in comparison with the state of the art, we carry out a calculation of 50 eigenpairs of $U$ with eigenvalues closest to 1 using (i) full diagonalization of $U$ using the `zgeev` Routine from MKL, (ii) shift invert diagonalization based on the LU decomposition of $U - \mathbb{1}$ using the `zgetrf` Routine from MKL in combination with `arpack`'s Arnoldi algorithm (iii) our new geometric sum filtered diagonalization using `arpack`'s Arnoldi algorithm.

We measure the total runtime of the calculation as well as the memory footprint and show the results in Tab. 1 and Fig. 3. We also calculate the maximal residue $\max_{n=1}^{50} \|U|n\rangle - \omega_n |n\rangle\|_2$ to check the quality of the obtained eigenpairs. Fig. 3 reveals that the scaling of all techniques is exponential in system size $L$ due to the nature of the problem. However, with a fixed runtime, geometric sum filter diagonalization yields eigenpairs of systems about 4 sites larger, i.e. for Hilbert spaces about 16 times larger. Due to the very small memory footprint, system sizes up to about $L = 20$ are therefore reachable, which would require about 50 TiB of memory with shift-invert. Despite the use of quite large polynomial orders, the algorithm is stable

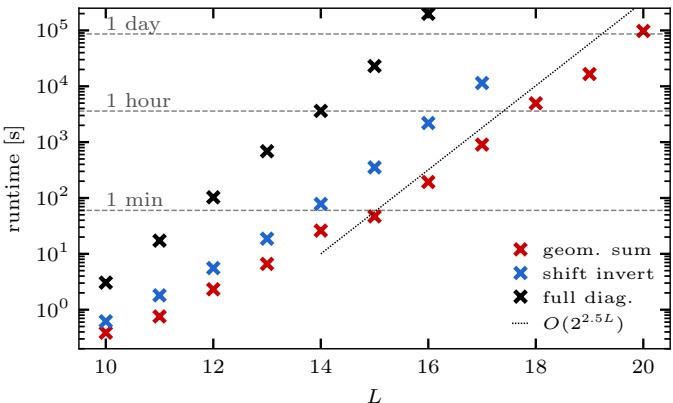

Figure 3: Scaling of total runtime for the calculation of $n_{\text{ev}} = 50$ eigenpairs close to $z_{\text{tgt}} = 1$ for different methods. The data is the same as in Tab. 1. The dotted line indicates the expected theoretical scaling $O(2^{2.5L})$ of the method.

and yields eigenpairs of excellent quality with residuals about one order of magnitude smaller than obtained from full diagonalization. As an example of how the obtained eigenstates of the unitary $U$ can be investigated, we show in Fig. 4 the entanglement entropy as a function of the subsystem size $L_A$. For this, we cut the system into a left half (subsystem $A$) consisting of $L_A$ qubits, and a right half (subsystem $B$) with $L - L_A$ qubits. Due to the open boundaries we use in this system, there is only one cut between the two subsystems. For each eigenstate $|\psi\rangle$, we can then calculate the entanglement entropy between the two subsystems, given by

$$S_A = -\text{Tr}\rho_A \ln \rho_A = -\sum_i s_i^2 \ln s_i^2, \quad \rho_A = \text{Tr}_B |\psi\rangle\langle\psi|. \tag{11}$$

Here, $s_i^2$ are the eigenvalues of the reduced density matrix $\rho_A$ of subsystem $A$, which can be calculated also by determining the singular values $s_i$ of the wave function reshaped as a matrix $\psi_{2^{L_A} \times 2^{L-L_A}}$, where the number of rows and columns reflect the dimensions of the Hilbert spaces of the respective subsystems. It is clear in Fig. 4 that the entanglement entropy follows a *volume law*, it is extensive as a function of subsystem size, almost up to $L_A = L$, at which point the entropy decreases due to the symmetry $S_A = S_B$. This clean scaling is expected, since our circuit (1) is designed to be ergodic. Since we are dealing with a Floquet system, all eigenstates are "infinite temperature" states, with *volume law* entanglement. This is why this system is not amenable to tensor network techniques for calculating exact eigenstates, since the required bond dimension would scale exponentially with system size. We can go one step further and compare these results quantitatively to the expected entanglement entropy for a cut of the system into two parts of size $L_A$ and $L_B$ ($L_A + L_B = L$) for random pure states. Page [56] conjectured this entropy to be for $L_A \leq L_B$

$$S_A = \sum_{k=2^{L_B}+1}^{2^L} \frac{1}{k} - \frac{2^{L_A} - 1}{2^{L_B+1}}, \tag{12}$$

and this was later proven to be correct [57, 58].

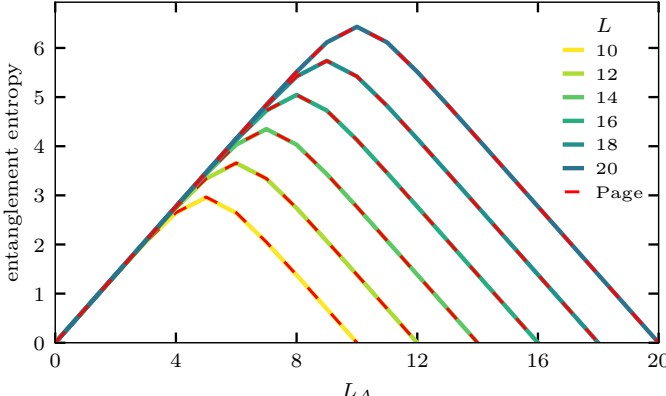

Figure 4: Entanglement entropy as a function of subsystem size $L_A$ for eigenstates of the unitary (1) for different system sizes $L = 10 \ldots 20$. For each curve, a different realization of the random circuit was generated and the mean over 50 eigenstates was taken. The variance over eigenstates is very small (of the order of $10^{-4}$ and smaller). The dashed red lines show the expected entanglement entropy for infinite temperature pure states [56].

The red dashed lines in Fig. 4 show the expected entropy (12), revealing a perfect match. This confirms the expectation, that in the random circuit model (1), the eigenstates are maximally chaotic.

# 6 Conclusion

We have shown that the geometric sum is an effective polynomial filter to obtain interior eigenpairs of local Floquet unitary operators. Due to the locality, an efficient matrix vector product $U|\psi\rangle$ can be defined and the geometric sum can be efficiently applied to any wave function. This allows the application of the implicitly restarted Arnoldi algorithm for finding eigenpairs closest to an arbitrary target on the complex unit circle.

Although the overall exponential scaling of the problem remains, the method has a moderate memory footprint compared to full diagonalization and shift-invert, and is roughly one order of magnitude faster than shift-invert, making systems of $L \geq 20$ qubits accessible.

We note that a large fraction of the runtime of the algorithm is spent in the matrix vector product due to the high order of the polynomial, and that runtimes can be reduced significantly by optimizing it.

# Acknowledgments

Comments from Luis Colmenarez are gratefully acknowledged. This work was in part supported by the Deutsche Forschungsgemeinschaft through SFB 1143 (project-id 247310070). This is an open data publication and the raw data as well as codes to generate the figures in this article are available in Ref. [59].

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

| method | $L$ | time [s] | mem. [GiB] | max res. |
|---|---|---|---|---|
| geom. sum | 10 | 0.3 | 0.167 | $4.2 \times 10^{-15}$ |
| geom. sum | 11 | 0.7 | 0.224 | $2.3 \times 10^{-15}$ |
| geom. sum | 12 | 2.3 | 0.265 | $1.9 \times 10^{-15}$ |
| geom. sum | 13 | 6.6 | 0.288 | $2.8 \times 10^{-15}$ |
| geom. sum | 14 | 26.0 | 0.369 | $2.5 \times 10^{-15}$ |
| geom. sum | 15 | 46.8 | 0.434 | $2.5 \times 10^{-15}$ |
| geom. sum | 16 | 193.3 | 0.861 | $2.5 \times 10^{-15}$ |
| geom. sum | 17 | 903.6 | 1.802 | $2.8 \times 10^{-15}$ |
| geom. sum | 18 | 4965.2 | 4.601 | $3.4 \times 10^{-15}$ |
| geom. sum | 19 | 16 579.0 | 12.081 | $3.7 \times 10^{-15}$ |
| geom. sum | 20 | 97 706.9 | 33.323 | $4.3 \times 10^{-15}$ |
| full diag. | 10 | 3.0 | 0.289 | $1.9 \times 10^{-14}$ |
| full diag. | 11 | 17.2 | 0.556 | $1.9 \times 10^{-14}$ |
| full diag. | 12 | 102.8 | 1.341 | $2.8 \times 10^{-14}$ |
| full diag. | 13 | 687.2 | 4.476 | $4.1 \times 10^{-14}$ |
| full diag. | 14 | 3622.3 | 16.622 | $4.0 \times 10^{-14}$ |
| full diag. | 15 | 23 025.9 | 64.581 | $5.2 \times 10^{-14}$ |
| full diag. | 16 | 198 568.0 | 192.614 | $6.8 \times 10^{-14}$ |
| shift invert | 10 | 0.6 | 0.264 | $5.7 \times 10^{-15}$ |
| shift invert | 11 | 1.8 | 0.454 | $6.2 \times 10^{-15}$ |
| shift invert | 12 | 5.5 | 0.996 | $9.0 \times 10^{-15}$ |
| shift invert | 13 | 18.6 | 3.25 | $1.7 \times 10^{-14}$ |
| shift invert | 14 | 78.3 | 12.263 | $1.4 \times 10^{-14}$ |
| shift invert | 15 | 352.9 | 48.19 | $2.7 \times 10^{-14}$ |
| shift invert | 16 | 2197.3 | 192.248 | $3.6 \times 10^{-14}$ |
| shift invert | 17 | 11 433.5 | 768.247 | $3.8 \times 10^{-14}$ |

Table 1: Average runtime and memory consumption for the calculation of $n_{\mathrm{ev}} = 50$ eigenpairs close to $z_{\mathrm{tgt}} = 1$ of our proposed geometric sum polynomial filter method with $n_{\mathrm{cv}} = \lfloor 2^{L/2+1} \rfloor$ and the optimal polynomial order $k = 0.8 \cdot 2^{L+1}/n_{\mathrm{cv}}$ (cf. Fig. 2) in comparison with full diagonalization using MKL's `zgeev`, as well as a custom shift invert implementation using MKL's `zgetrf`. Runtimes were measured using 16 cores of an AMD EPYC 7H12 2.6 GHz CPU. Memory usage is estimated from the maximum resident set size (max RSS). Memory footprints are only indicative since our simple benchmark codes were not optimized for memory usage.