# Peer review of "Polynomial filter diagonalization of large Floquet unitary operators"

_SciPost Physics_

## Round 1 · Referee Report · Anonymous (Referee 1) · 2021-4-26

Strengths

1-Useful extension of standard ED methods for (non-)Hermitian matrices to unitary matrices, which can (as argued) be immediately relevant for Floquet systems.

2-The presented methods are clearly correct and their advantages over other methods are supported by strong numerical evidence.

3-The main ideas are clearly presented.

Weaknesses

1-The text may be a bit technical and to concise for the general reader.

2-No data is provided. No code is provided – not even pseudo-code.

Report

The paper introduces an adaptation of general ED methods to unitary matrices. In particular, a polynomial function is introduced that maps a complex number on the unit circle to a general complex number, whose norm depends on how close the initial number is to a specified target (on the unit circle). The polynomial in question is a finite geometric sum, which could also be understood as a sort of truncated delta function around the target, hence the close values are strongly enhanced. This function allows to compute the eigenvalues and eigenvectors of a unitary matrix close to a set target using only matrix-vector multiplications (due to the polynomial form). Since relevant unitary matrices will be dense in general, this can be far more efficient than shift-invert methods as argued and demonstrated in the paper. In between, it is also described how an efficient matrix-vector multiplication of a unitary based on a “brickwork” circuit can be performed.

After introducing these concepts, the paper mostly presents numerical evidence of the superiority of the polynomial method as opposed to the shift-invert method and full diagonalization. For this, 50 eigenpairs close to 1 are computed for different sizes and with different parameters. The evidence shows clearly that there is an advantage in terms of runtime and memory requirements. Given that polynomial methods are well established also for general systems, this is really not too surprising, hence there is no reason to doubt the reported findings.

Therefore, the question of publication boils down to the question of whether the results offer a significant numerical advantage and whether they are interesting to the general reader. One could argue that it is more or less a straightforward application of known polynomial methods to unitary matrices – this is also the feeling I got when first reading the text. However, given that unitary matrices are very important, not only in Floquet but also in Quantum Physics generally, and given that work on similar methods for Hermitian systems has been published and is widely known, I believe that the paper should be published in SciPost Physics.

However, I feel that some changes (apart from using the SciPost layout) should be made as listed below in the requested changes section. A change that I do not request, but would suggest the author to consider, would be to add some discussion of how computing eigenpairs for larger sizes could be useful. I think that already a short discussion of concepts like level statistics and their meaning in the study of phase transitions or the study of heating in Floquet systems could increase the readability and usefulness for the general reader significantly.

Acceptance criteria (https://scipost.org/SciPostPhys/about#criteria):
In my opinion (see the discussion earlier) the paper meets the SciPost expectation

“3. Open a new pathway in an existing or a new research direction, with clear potential for multipronged follow-up work;”.
.

It also meets the general acceptance criteria except for

“5. Provide (directly in appendices, or via links to external repositories) all reproducibility-enabling resources: explicit details of experimental protocols, datasets and processing methods, processed data and code snippets used to produce figures, etc.;”

While I am confident that the results can be reproduced in principle given the paper, no code for the computations or figures and no data is provided.

Requested changes

0-(formal) Provide the necessary code and data to fulfill the reproducibility requirement. Use the SciPost layout.

1-The expressions in eq. 6 and 7 should be discussed in a bit more detail – at least in an Appendix. For instance the functional form $g_{k}(e^{i \theta}) $should be provided, such that one sees the equation of the “apple”. Also the derivation of the mapping of an arc to the “apple” should be be given.

2-Concerning Fig. 2: I find the color scheme here rather confusing. Not only is the scale different for every plot, so that one can not compare the sizes based on color (to be fair this is done in Table 1 and Fig. 3), the relative ranges are also completely different. For instance in the top figure the numbers range from (presumably) 1 to around 11 (?), so about a factor of 10, while in the bottom they range from 600 (?) to 1250 (?), so about a factor of 2. As a result there seems to be a lot of variation in the color of of the bottom plot, while the top plot seems almost fully blueish. Thus the plots look visually different but there is no real meaning behind it, or it is not explained well – in any case there should be some changes in the figure or discussion thereof.

3-A clarification as to the exact definition of the random circuit might be useful: is every unitary drawn from a circular ensemble or is there a finite set of unitaries and they are chosen at random? Is there an averaging involved in the benchmarks?

4-Is the “fast” matrix-vector multiplication used in all benchmarks (if applicable)?

5-Curiosity: Do you know why the residuals for full diagonalization in Tab. 1 are so bad (compared to the other methods)?

6-Is there an insight as to why one gains around four sites, or is this a purely empirical observation?

7-In the last sentence of the conclusion possible optimizations are hinted at, perhaps at least a sketch of how these could be achieved could be given here.

8-Finally, the references should be extended. While the bibliography is complete in the sense that all statements are referenced, it seems that there are extensive references for some topics like discrete time crystals (13 out of 40 – by the way references 21-23 are arguably not related to time crystals but rather focus on prethermalization and energy absorption in generic systems),
while for other topics only a single reference is used with more well known works being available. As an example, [1] is used as a reference for heating to an infinite temperature state in generic Floquet systems, but 10.1103/PhysRevX.4.041048 is a well known source for the same statement. Other works I can see fitting (off the top of my head) are: 10.1103/PhysRevLett.116.120401, 10.1080/00018732.2016.1198134, 10.1103/RevModPhys.89.011004, 10.1080/00018732.2015.1055918, 10.1103/RevModPhys.91.021001.

  • validity: top
  • significance: high
  • originality: good
  • clarity: top
  • formatting: good
  • grammar: excellent

Author:  David J. Luitz  on 2021-06-16  [id 1511]

(in reply to Report 1 on 2021-04-26)

I appreciate the recommendation for publication in SciPost Physics and thank the referee for the positive assessment and the constructive comments, which I address below.

  1. The new version now uses the SciPost layout. I have published all plotting scripts and data necessary to generate the figures of the manuscript here: https://edmond.mpdl.mpg.de/imeji/collection/PgO6Q2vWIVXjxJ33 Once the paper will be in press, I will generate a DOI for this dataset and include a reference in the manuscript.

  2. I have added a more detailed discussion to explain the mapping of the arc to the apple to make the logic more clear.

  3. I agree that the choice of color range was not optimal. I have changed this, but notice that for smaller sizes the variance in runtimes is not very large, this only grows for larger sizes. The point of this figure is to reveal the pronounced minimum in the region where the polynomial order equals $0.8\cdot 2^L/n_{\mathrm cv}$, as expected from the arguments on the arc to apple mapping.

  4. I have added some details on the model. No averaging was performed, but each run of the simulation (e.g. each color point in Fig. 2) used a different realization of the circuit. For benchmarking the exact realization is of little importance.

  5. Yes, the "fast" matrix vector product was used in the entire work.

  6. This is also not entirely clear to me. It seems that nonhermitian diagonalization routines are not extremely optimized both in runtime and precision. I suspect that the loss of precision stems from the fact that no assumption of an orthogonal eigenbasis is included in these algorithms, since they are designed for general matrices. I am not aware of optimized general eigensolvers for unitary matrices with orthogonal eigenbasis.

  7. This is purely numerical, but I think the gain is partly due to a tradeoff: instead of full diagonalization, we only ask for a part of the spectrum, which yields a speedup. This argument is however also true for shift-invert diagonalization, which performs worse both in terms of runtime and memory.

  8. This depends very much on the problem, so I prefer to keep this comment general. One could for example imagine outsourcing this part of the computation to a GPU, although one has to check the tradeoff of gained performance vs the cost of data transfer to the device.

  9. I have added the mentioned references and included further references.

---

## Round 1 · Referee Report · Anonymous (Referee 2) · 2021-4-28

Strengths

  • The manuscript proposes a new diagonalization scheme for Floquet operators

  • The proposed method is convincingly demonstrated to have an advantage over existing exact diagonalization schemes, both in terms of time and memory

Weaknesses

  • No new physics, no results regarding the physical properties of the studied system

  • The complexity of the algorithm is not discussed

Report

The manuscript "Polynomial filter diagonalization of large Floquet unitary operators" introduces a method of exact diagonalization of unitary Floquet operators. The method relies on an observation that one can formulate an efficient multiplication of vectors by Floquet operators for physically interesting systems. A combination of a polynomial spectral transformation and Arnoldi algorithm that utilizes the efficient matrix-vector product is then used to calculate eigenvalues of the Floquet operator.
The manuscript shows that the proposed method indeed provides eigenvectors of the considered Floquet operator and convincingly argues that the proposed method significantly outperforms both full exact diagonalization and shift and invert method. The Floquet systems play a major role in our understanding of non-equilibrium dynamics. Hence, the paper fulfils the two expectations of submissions to SciPost Physics: i) it details an important computational discovery, ii) opens a new pathway in an existing research direction. As such, it should be published in SciPost Physics. However, I recommend a revision of the manuscript according to the remarks specified below.

Requested changes

  1. The paper presents no results on the considered physical system. In fact it is even not clear what precisely is the considered system: are the two-site unitaries drawn with the Haar measure? While the aim of the paper is to introduce the method, including results showing some basic physical properties of the calculated eigenstates (for instance the system size dependence of entanglement entropy) would widen the scope of the paper.

  2. From the present manuscript it is not clear what is the complexity of the proposed method. A more thorough analysis of the scaling of the various steps of the algorithm with matrix size should be included. Those scalings could then be compared with results shown in Fig. 3.

  3. Could the efficiency of the method be improved by varying the coefficients of the polynomial in the spectral transformation (6)?

  4. Can investigations of Floquet systems of size L=20 bring new physics as compared to systems of size L=15 accessible to full exact diagonalization? I believe that referencing the recent debate about many-body localization transition [Phys. Rev. E 102, 062144 (2020), EPL 128, 67003 (2020), Phys. Rev. Lett. 124, 186601 (2020), Annals of Physics 427, 168415 (2021), etc.] would illustrate well the importance of finite size effects in non-equilibrium phenomena.

  • validity: high
  • significance: high
  • originality: high
  • clarity: good
  • formatting: perfect
  • grammar: perfect

Author:  David J. Luitz  on 2021-06-16  [id 1510]

(in reply to Report 2 on 2021-04-28)
Category:
answer to question

I thank the referee for constructive comments and for the recommendation for publication in SciPost Physics. The resubmitted version implements changes suggested by both referees and I comment on the requested changes below. 1. Indeed the aim of this paper is to present a new method, and the focus is clearly not on the physics of the benchmark system. This being said, I have added some details on the random circuit. Yes, the unitaries are sampled from the Haar measure. I have adopted the suggestion of the referee, showing the entanglement scaling of eigenstates of the circuit for all possible bipartitions in comparison to the prediction for random wave functions by Page, yielding perfect agreement as expected. This result also illustrates the power of the method, since these wave functions can not be efficiently represented by matrix product states and hence the discussed method is currently the only one giving access to eigenstates of such large systems.

  1. The complexity of the method is dominated by the complexity of the matrix vector product. To get an idea, I have added an estimate of the complexity for the circuit matrix vector product introduced in this paper, yielding an asymptotic scaling of the order of $O(2^{2.5 L})$. This theoretical scaling is however not perfectly seen in the benchmarks, presumably due to significant CPU specific effects and optimizations, where e.g. cache sizes and OpenMP efficiency still play a role. The observed scaling seems slightly better for the accessible system sizes.

  2. I am convinced that the proposed polynomial is the optimal choice in the sense that it maximizes the absolute value of the mapped target arc and minimzes it outside the target region. I have found this by numerical optimization, but suspect that it can be proved. I have added this conjecture to the paper.

  3. I fully agree with the referee that this method can and should be used to study the Floquet MBL transition, and since Floquet models are "cleaner" due to their flat density of states, one can expect to make some progress here with the extended range of available system sizes. I have added the suggested references to the paper and a comment to this effect.

---

## Editorial Decision

resubmitted